# Accelerometer compared with questionnaire measures of physical activity in relation to body size and composition: a large cross-sectional analysis of UK Biobank

Wenji Guo, Timothy J Key, Gillian K Reeves

Nuffield Department of Population Health, University of Oxford, Oxford, UK

**Correspondence to**
Dr Wenji Guo; wguo8@jhmi.edu

## ABSTRACT

**Objectives** Previous studies of the association between physical activity and adiposity are largely based on physical activity and body mass index (BMI) from questionnaires, which are prone to inaccurate and biased reporting. We assessed the associations of accelerometer-measured and questionnaire-measured physical activity with BMI, waist circumference and body fat per cent measured by bioelectrical impedance and dual-energy X-ray absorptiometry (DXA).

**Design** Cross-sectional analysis of UK Biobank participants.

**Setting** UK Biobank assessment centres.

**Participants** 78 947 UK Biobank participants (35 955 men and 42 992 women) aged 40–70 at recruitment, who had physical activity measured by both questionnaire and accelerometer.

**Main outcome measures** BMI, waist circumference and body fat per cent measured by bioelectrical impedance.

**Results** Greater physical activity was associated with lower adiposity. Women in the top 10th of accelerometer-measured physical activity had a 4.8 (95% CI 4.6 to 5.0) kg/m$^2$ lower BMI, 8.1% (95% CI 7.8% to 8.3%) lower body fat per cent and 11.9 (95% CI 11.4 to 12.4) cm lower waist circumference. Women in the top 10th of questionnaire-measured physical activity had a 2.5 (95% CI 2.3 to 2.7) kg/m$^2$ lower BMI, 4.3% (95% CI 4.0% to 4.5%) lower body fat per cent and 6.4 (95% CI 5.9 to 6.9) cm lower waist circumference, compared with women in the bottom 10th. The patterns were similar in men and also similar to body fat per cent measured by DXA compared with impedance.

**Conclusion** Our findings of approximately twofold stronger associations between physical activity and adiposity with objectively measured than with self-reported physical activity emphasise the need to incorporate objective measures in future studies.

## Strengths and limitations of this study

► This study uses data on physical activity objectively measured by accelerometer rather than only self-reported data from questionnaires, which may be prone to inaccurate and potentially biased reporting.

► This study is by far the largest study to examine associations of objectively measured physical activity and self-reported physical activity with various measures of adiposity, including body fat per cent assessed by bioelectrical impedance and dual-energy X-ray absorptiometry.

► Due to the cross-sectional nature of this study, we cannot assess to what extent physical activity is causally related to adiposity.

activity is generally accepted to be important for the prevention of weight gain, achievement of modest weight loss and prevention of weight regain after weight loss,[3] randomised controlled trials have shown inconsistent results, perhaps partly due to limited duration of interventions and difficulty in long-term adherence to exercise regimens,[4] and previous large-scale observational studies are mostly based on self-reported physical activity from questionnaires, which are prone to both inaccurate reporting and reporting bias.[5]

Prior studies have demonstrated low to moderate correlation between self-reported and objective accelerometer measures of physical activity.[6 7] Self-reported and accelerometer-measured physical activity capture different aspects of physical activity with limitations unique to each.[7] However, research methods using more objective measures of physical activity, along with more detailed measures of body fat, are needed to reduce measurement error and more accurately characterise the association between physical activity and adiposity.

## INTRODUCTION

The prevalence of overweight and obesity is high worldwide and is associated with increased risk of various conditions including heart disease, stroke, hypertension, diabetes and some cancers.[1 2] Although physical

We examined the association between physical activity and adiposity, with accelerometer-measured compared with self-reported physical activity in nearly 80 000 participants. These associations were assessed using various measures of adiposity, including body mass index (BMI), waist circumference and body fat per cent measured by both bioelectrical impedance and dual-energy X-ray absorptiometry (DXA). We also explored how the associations vary by age.

## METHODS

### Data source

Data were obtained from UK Biobank. Details of UK Biobank design, rationale and survey methods have been described elsewhere.[8] Information on data available and access procedures are described on the study website (http://www.ukbiobank.ac.uk/).

### Study participants

The complete UK Biobank dataset includes 502 617 UK adults (229 164 men and 273 453 women) between 40 and 70 years of age at recruitment during 2006–2010. During the baseline assessment centre visit, participants completed a touch screen questionnaire which included questions on sociodemographics, lifestyle, health and medical history, and sex-specific factors. The present study was restricted to participants with available accelerometer data (n=103 705). Participants were excluded if they did not have at least 72 hours of data and also data in each 1-hour period of the 24 hours cycle across multiple days (n=6995). Participants were also excluded if they had insufficient data for calibration (n=4). Participants who had missing data on any of the physical activity variables used in our analyses were excluded (n=15 999). Participants who reported physical activity greater than an average of 16 hours per day (n=620) were also excluded as recommended by the International Physical Activity Questionnaire (IPAQ) scoring guidelines, which can be accessed at file:///H:/Downloads/GuidelinesforDataProcessingandAnalysisoftheInternationalPhysicalActivityQuestionnaireIPAQShortandLongForms.pdf. Finally, participants with missing data on BMI (n=146), body fat per cent (n=988) and waist circumference (n=6) were excluded. The analyses included 35 955 men and 42 992 women (online supplementary figure 1).

### Self-reported physical activity

Physical activity questions from the baseline questionnaire captured the frequency and duration of three intensities of activity (walking, moderate and vigorous). Participants were asked how many days per week they typically engaged in each category of activity. For each category in which an answer of 1 or more days was given, the participant was subsequently asked the number of minutes on average spent on the activity per day. Questions were adapted from the IPAQ, a validated survey instrument,[9] and are listed in online supplementary table 1. Metabolic equivalents (METs) were used to quantify physical activity; 1 MET is expended by sitting quietly for 1 hour, and the MET value reflects the ratio of energy expended per kilogram of body weight per hour to that expended when sitting quietly.[10] The number of minutes per day engaged in each level of activity was multiplied by the respective MET score for the corresponding level of activity (3.3 for walking, 4.0 for moderate physical activity and 8.0 for vigorous physical activity).[11] MET-minutes per day were converted to MET-hours per week. The total amount of METs was calculated by summing total METs from the walking, moderate and vigorous activity levels. Following IPAQ scoring guidelines, physical activity of less than 10 min per day for any category was recoded to 0.

### Accelerometer-measured physical activity

A total of 236 519 participants, all of whom had provided a valid email address, were invited to participate in a 7-day accelerometer study between February 2013 and December 2015 (on average, approximately 5.5 years after recruitment when baseline physical activity was self-reported). Starting in June 2013, participants were sent wrist-worn triaxial accelerometers (Axivity AX3, Newcastle upon Tyne, UK) that were programmed to capture three-dimensional acceleration data at 100 Hz with a dynamic range of ±8 g. Participants were also given instructions to wear the accelerometer on their dominant wrist continuously for 7 days and then to send the device to the coordinating centre using the provided prepaid envelope. Further details on data collection, processing and analysis can be found elsewhere.[12] We used the 'overall acceleration average' variable (data field 90012) in the present analyses.

### Anthropometry and body composition

At the UK Biobank baseline interview, trained staff measured standing height using the Seca 202 device (Seca, Hamburg, Germany). BMI was calculated by dividing weight (kg) by the square of standing height (m$^2$). The Wessex non-stretchable sprung tape measure (Wessex, UK) was used to measure waist circumference at the level of the umbilicus. The Tanita BC-418MA body composition analyzer (Tanita, Tokyo, Japan) was used to measure body fat per cent using bioelectrical impedance. DXA was used to measure fat per cent on a subset of 2457 participants included in the present study, beginning in 2014 using the GE-Lunar iDXA (GE Healthcare, Chicago, Illinois, USA).

### Statistical analyses

Baseline characteristics were summarised by physical activity (least active fifth, most active fifth and overall) separately for men and women. Since self-reported physical activity was not normally distributed, Spearman's correlation coefficients were used to measure the strength of correlations between self-reported and accelerometer-measured physical activity in the overall population and in subgroups based on sociodemographic characteristics.

Self-reported and accelerometer-measured physical activity were categorised into 10ths and the median value within each category of physical activity is shown in the figures. The associations of physical activity and adiposity measures were examined using multivariable linear regression, separately in men and women. Analyses comparing the association of accelerometer-measured physical activity with body fat per cent, measured by bioelectrical impedance and DXA were restricted to participants with both measures. Likelihood ratio tests were used to assess whether the associations between physical activity and adiposity were modified by age (<55 years or 55+ years), separately for self-reported and accelerometer-measured physical activity.

Covariates were determined a priori and were 5-year age at recruitment categories, socioeconomic status as indicated by fifths of Townsend Deprivation Index,[13] educational qualifications, employment status, smoking status (never, previous, current) and alcohol intake frequency. Analyses in women were additionally adjusted for parity (nulliparous, 1, 2, 3, 4 or more births) and hormone replacement therapy use (never, previous, current). As a covariate, educational qualification was grouped into the following categories: vocational qualifications, national examinations at age 16 (Ordinary-levels, General Certificate of Secondary Education, Certificate of Secondary Education or equivalent), optional national examinations at ages 17–18 years (A levels, AS levels or equivalent) and college or university degree. Employment status was categorised as paid or self-employed, retired, looking after a home and/or family, unemployed, doing unpaid or voluntary work, unable to work due to sickness or disability and student. Alcohol intake was categorised as never, special occasions only, 1–3 times a month, 1–2 times a week, 3–4 times a week and daily or almost daily.

Missing data were grouped in a separate unknown category for each covariate. There were less than 1% missing data for all covariates except for educational qualifications (7.4% missing data). To assess the impact of missing values, a sensitivity analysis restricted to participants with known values for all covariates was conducted. We also conducted sensitivity analyses to assess the impact of excluding participants who reported long-term illness, disability or infirmity, participants who reported fair or poor health rather than excellent or good health, and participants whose jobs usually or always required heavy manual work. Analyses were conducted using STATA, V.15.0 (StataCorp).

## Patient and public involvement

This study did not involve patients and the public.

## Results

Characteristics of the study population by the least active and most active fifth of accelerometer-measured physical activity are shown in table 1.

Mean accelerometer-measured physical activity was 27.6 (SD 8.7) milli-gravity in men and 28.7 (SD 8.0) milli-gravity in women. The most active participants were on average younger and had lower values for all body size and composition variables. They were more likely to have a college or university degree, be employed rather than retired, have an active job and consume alcohol at least weekly. The least active participants were more likely to be ever smokers and were also more likely to have a long-standing illness or disability. The correlation between questionnaire and accelerometer-measured physical activity, recorded on average 5.5 years later, was 0.24 (95% CI 0.23 to 0.25) in men (online supplementary table 2) and 0.22 (95% CI 0.21 to 0.23) in women (online supplementary table 3). The correlations were comparatively higher in participants who were younger and in participants who had lower BMI. The correlations were lower among participants who reported that their job usually or always involved heavy manual work and/or mainly walking or standing.

The inverse associations between physical activity and all measures of adiposity were linear and approximately twofold larger in models that used accelerometer measured rather than self-reported physical activity. Since there was heterogeneity in the associations between both self-reported and accelerometer-measured physical activity and adiposity by sex (p<0.001), separate analyses were performed in men and women. The mean differences in BMI and body fat per cent were greater in women compared with men. Comparing the top to bottom 10th of accelerometer-measured physical activity, the difference in BMI was 4.8 (95% CI 4.6 to 5.0) kg/m$^2$ in women and 3.6 (95% CI 3.4 to 3.8) kg/m$^2$ in men (figure 1, online supplementary table 4).

Women in the top 10th of accelerometer-measured physical activity had an 8.1% (95% CI 7.8% to 8.3%) lower body fat per cent while women in the top 10th of self-reported physical activity had a 4.3% (95% CI 4.0% to 4.5%) lower body fat per cent, compared with those in the bottom 10th of physical activity. Men in the top 10th of accelerometer-measured physical activity had a 6.0% (95% CI 5.7% to 6.2%) lower body fat per cent while men in the top 10th of self-reported physical activity had a 3.6% (95% CI 3.3% to 3.8%) lower body fat per cent, compared with those in the bottom 10th (figure 1, online supplementary table 4).

Associations between physical activity and waist circumference were of similar magnitude in men and women, with an approximately twofold larger inverse association between waist circumference and physical activity when measured by accelerometer rather than questionnaire (figure 1, online supplementary table 4).

The results of sensitivity analyses excluding participants who had any missing values, reported a long-term illness or disability, reported a health rating worse than 'good', or whose jobs usually or always required heavy manual work did not materially differ from the main findings.

Figure 2 and online supplementary table 5 show the associations between accelerometer-measured physical activity and bioelectrical impedance-measured body fat

**Table 1** Characteristics of the UK Biobank study population, according to fifths of accelerometer-measured physical activity

| | Least active men <20.8 milli-gravity | Most active men ≥33.7 milli-gravity | All men | Least active women <22.2 milli-gravity | Most active women ≥34.6 milli-gravity | All women |
|---|---|---|---|---|---|---|
| No of participants | 7202 | 7186 | 35 955 | 8606 | 8595 | 42 992 |
| Age at recruitment (years), mean (SD) | 59.7 (7.0) | 53.4 (7.7) | 56.7 (7.9) | 58.0 (7.4) | 52.6 (7.4) | 55.3 (7.7) |
| Lowest fifth of socioeconomic status | 1520 (21.1%) | 1351 (18.8%) | 6800 (18.9%) | 1897 (22.0%) | 1699 (19.8%) | 8744 (20.3%) |
| Accelerometer-measured physical activity (milli-gravity), mean (SD) | 17.5 (2.6) | 40.5 (7.8) | 27.6 (8.7) | 18.9 (2.7) | 40.6 (6.3) | 28.7 (8.0) |
| Self-reported physical activity (MET-hours/week), median (IQR) | 20.7 (9.0–42.6) | 44.2 (23.7–80.9) | 29.9 (14.2–58.1) | 21.3 (9.9–41.7) | 40.2 (21.8–73.2) | 29.3 (14.4–55.3) |
| Height (cm), mean (SD) | 176.3 (6.8) | 176.4 (6.6) | 176.5 (6.6) | 163.2 (6.3) | 163.7 (6.1) | 163.5 (6.2) |
| Weight (kg), mean (SD) | 89.4 (15.4) | 80.8 (11.4) | 84.9 (13.5) | 75.5 (15.6) | 65.0 (10.3) | 69.9 (13.2) |
| BMI (kg/m$^2$), mean (SD) | 28.8 (4.6) | 25.9 (3.3) | 27.2 (4.0) | 28.3 (5.7) | 24.3 (3.7) | 26.2 (4.8) |
| Body fat per cent (%)*, mean (SD) | 27.0 (5.6) | 21.7 (5.4) | 24.4 (5.7) | 38.7 (6.6) | 31.7 (6.4) | 35.3 (6.8) |
| Waist circumference (cm), mean (SD) | 100.1 (11.7) | 90.9 (9.3) | 95.4 (10.8) | 87.9 (13.3) | 77.6 (9.5) | 82.4 (11.7) |
| College or university degree | 3018 (41.9%) | 3365 (46.8%) | 16 709 (46.5%) | 3586 (41.7%) | 4060 (47.2%) | 19 214 (44.7%) |
| Current employment status | | | | | | |
| Paid employment/self-employed | 3608 (50.1%) | 5420 (75.4%) | 22 942 (63.8%) | 4401 (51.1%) | 6101 (71.0%) | 26 693 (62.1%) |
| Retired | 3107 (43.1%) | 1451 (20.2%) | 11 361 (31.6%) | 3517 (40.9%) | 1591 (18.5%) | 12 710 (29.6%) |
| Other | 487 (6.8%) | 315 (4.4%) | 1652 (4.6%) | 688 (8.0%) | 903 (10.5%) | 3589 (8.3%) |
| Job involves mainly walking/standing† | 707 (19.6%) | 1742 (32.1%) | 5574 (24.3%) | 893 (20.3%) | 1926 (31.6%) | 6648 (24.9%) |
| Job involves heavy manual work‡ | 272 (7.5%) | 912 (16.8%) | 2335 (10.2%) | 170 (3.9%) | 576 (9.4%) | 1567 (5.9%) |
| Weekly or more frequent alcohol intake | 5545 (77.0%) | 5989 (83.3%) | 29 421 (81.8%) | 5295 (61.5%) | 6292 (73.2%) | 29 829 (69.4%) |
| Ever smoker | 3801 (52.8%) | 3126 (43.5%) | 16 964 (47.2%) | 3583 (41.6%) | 3212 (37.4%) | 16 936 (39.4%) |
| Long-standing illness or disability | 3089 (42.9%) | 1543 (21.5%) | 10 825 (30.1%) | 3145 (36.5%) | 1449 (16.9%) | 10 685 (24.9%) |

*Body fat per cent was measured by bioelectrical impedance.
†Participants who reported their work 'usually' or 'always' involved walking or standing for most of the time.
‡Participants who reported their work 'usually' or 'always' involved heavy manual or physical work for most of the time.
BMI, body mass index; MET, metabolic equivalent.

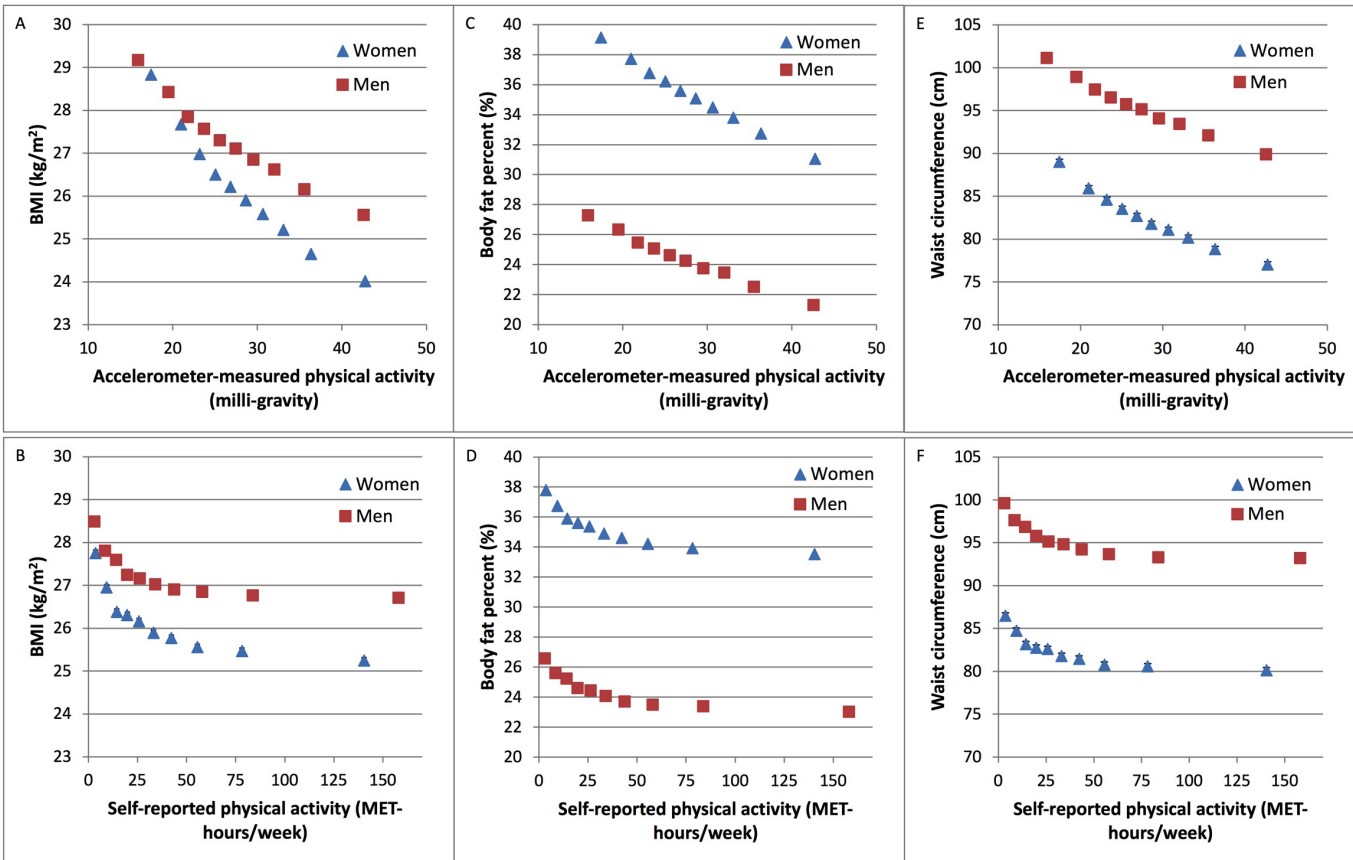

**Figure 1** Association of self-reported and accelerometer-measured physical activity with adiposity variables in UK Biobank. Association of (A) accelerometer-measured and (B) self-reported physical activity with BMI. Association of (C) accelerometer-measured and (D) self-reported physical activity with body fat per cent. Association of (E) accelerometer-measured and (F) self-reported physical activity with waist circumference. Physical activity was grouped into 10ths, separately in men and women. Adjusted geometric means (from linear regression models) for BMI, body fat per cent and waist circumference are plotted against the median value within each 10th of self-reported or accelerometer-measured physical activity. Adjusted geometric means are represented by squares for men and triangles for women. These analyses are stratified by age at recruitment, region of recruitment and socioeconomic status, and are adjusted for educational qualifications, employment status, smoking status and alcohol intake frequency. Analyses in women are additionally adjusted for parity and hormone replacement therapy use. The figure shows point estimates and 95% CIs. BMI, body mass index; MET, metabolic equivalent.

per cent at baseline (2006–2010) compared with body fat per cent measured by DXA starting in May 2014. Body fat per cent by impedance at baseline was lower than body fat per cent by DXA, measured on average 7 years later. For both measures, there was a linear dose–response association between physical activity and body fat per cent in both men and women. The inverse associations were stronger when body fat per cent was measured by DXA. Compared with the least active women, the most active women had an 8.8% (95% CI 7.7% to 10.0%) lower DXA-measured body fat per cent and a 7.0% (95% CI 5.9% to 8.1%) lower impedance-measured body fat per cent (figure 2 and online supplementary table 5). Associations between physical activity and measures of adiposity by age group are shown in figure 3 for men and figure 4 for women. For a given level of accelerometer-measured physical activity, the older participants (over age 55) had a slightly lower BMI but a higher body fat per cent compared with their younger counterparts. For women, there was a

heterogeneity by age in the association between self-reported physical activity and body fat per cent (p=0.03) but there was no heterogeneity by age when physical activity was measured by the accelerometer (p=0.27).

## DISCUSSION

In this large cross-sectional study of nearly 80 000 participants, we found that associations between physical activity and BMI, body fat per cent, and waist circumference were stronger when physical activity was measured by accelerometer compared with questionnaire self-reports. Body fat per cent measured by DXA at follow-up showed a slightly stronger association with physical activity compared with body fat per cent measured by bioelectrical impedance at baseline, but the overall pattern of association was similar. The correlation between accelerometer-measured and self-reported physical activity, recorded 5.5 years apart,

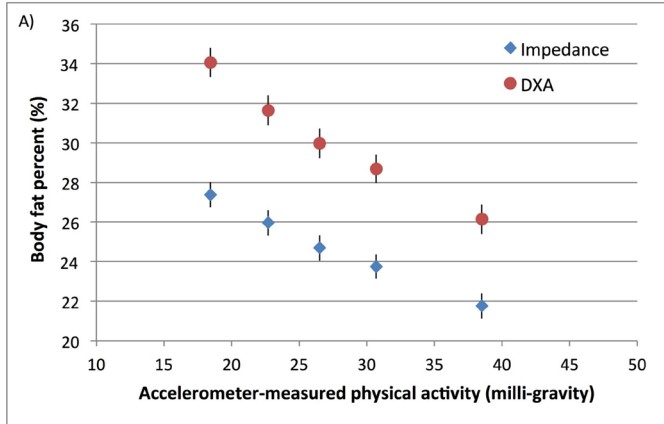

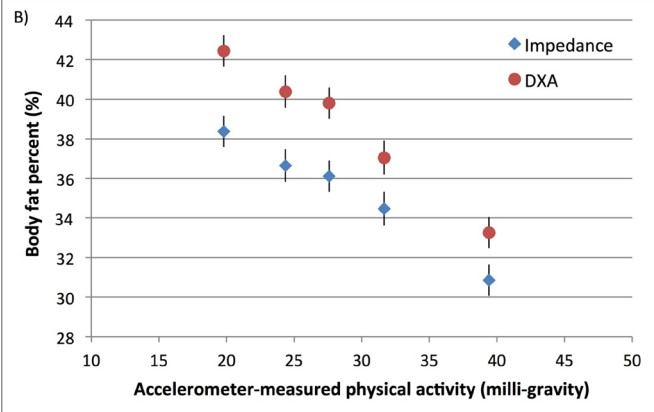

**Figure 2** Association of accelerometer-measured physical activity with body fat per cent measured by impedance and DXA in UK Biobank (A) men (n=1185) and (B) women (n=1272). Physical activity was grouped into fifths, separately in men and women. Adjusted geometric means (from linear regression models) for body fat per cent are plotted against the median value within each fifth of accelerometer-measured physical activity. Adjusted geometric means are represented by diamonds for body fat per cent measured by impedance and circles for body fat per cent measured by DXA. These analyses are restricted to participants with measures of body fat per cent by both impedance and DXA. Analyses are stratified by age at recruitment, region of recruitment and socioeconomic status, and are adjusted for educational qualifications, employment status, smoking status and alcohol intake frequency. Analyses in women are additionally adjusted for parity and hormone replacement therapy use. The figure shows point estimates and 95% CIs. DXA, dual-energy X-ray absorptiometry.

was lower in participants with higher BMI and in older participants.

There was a consistent dose–response relationship between physical activity and adiposity across the different measures of adiposity, which are highly correlated.[14] Our analyses based on accelerometer-measured physical activity suggest an approximately linear inverse association between physical activity and adiposity, with the most active participants having the lowest BMI, body fat per cent and waist circumference. In contrast, the analyses in the same participants based on self-reported physical activity

suggest a comparatively small further benefit of physical activity greater than 50 MET-hours a week on adiposity.

We have previously suggested that the steeper inverse association between physical activity and adiposity within the lower range of physical activity could be due to either a comparatively larger benefit of physical activity for those who are relatively inactive or measurement error from over-reporting of high physical activity.[14] The present analyses demonstrating an approximately linear dose–response relationship between accelerometer-measured physical activity and adiposity supports the latter explanation and further suggests that over-reporting of total physical activity contributed to the low overall correlation between self-reported and accelerometer-measured physical activity, although the time lag between these two measurements of physical activity may have also contributed to a low overall correlation coefficient. Although wrist accelerometer-measured physical activity also has limitations, such as a measuring movement of only one part of the body and the inability to reliably capture activities such as cycling,[7] it has the major advantage of eliminating both inaccurate reporting that leads to a random error as well as reporting bias that may vary by sociodemographic characteristics.

Measurement error in the self-reported data results in misclassification of individuals by physical activity status. We used the IPAQ short form data processing rules since the UK Biobank questionnaire did not comprehensively cover domain-specific activities, but it is still likely that lower intensity activities were under-reported and reported less accurately.[15] In contrast, the accelerometers were worn for 24 hours a day, over 7 days. Therefore, the lower correlation between self-reported and accelerometer-measured physical activity in older participants[16] and the heterogeneity by age seen only with the self-reported data may be explained by the observation that, in older adults, a greater proportion of physical activity is of lower intensity.[17]

Individuals with higher body fat per cent may report moderate and strenuous physical activity less accurately than leaner individuals, based on comparisons between self-reported physical activity and energy expenditure estimated from whole-room indirect calorimeter.[18] In agreement with some previous studies, we found that the correlation between physical activity measured by questionnaire and accelerometer-measured physical activity was greater for those with lower BMI.[7] This suggests that measurement error of self-reported physical activity may be greater in overweight and obese BMI groups.

We, like several prior studies, found stronger associations between accelerometer-measured physical activity and all measures of adiposity in women compared with men.[19–21] This may partly be due to the fact that, in the present study, men were on average objectively less physically active than women. Differences in fat metabolism may also play a role, with a greater proportion of energy derived from lipolysis during exercise in women compared with men.[21 22]

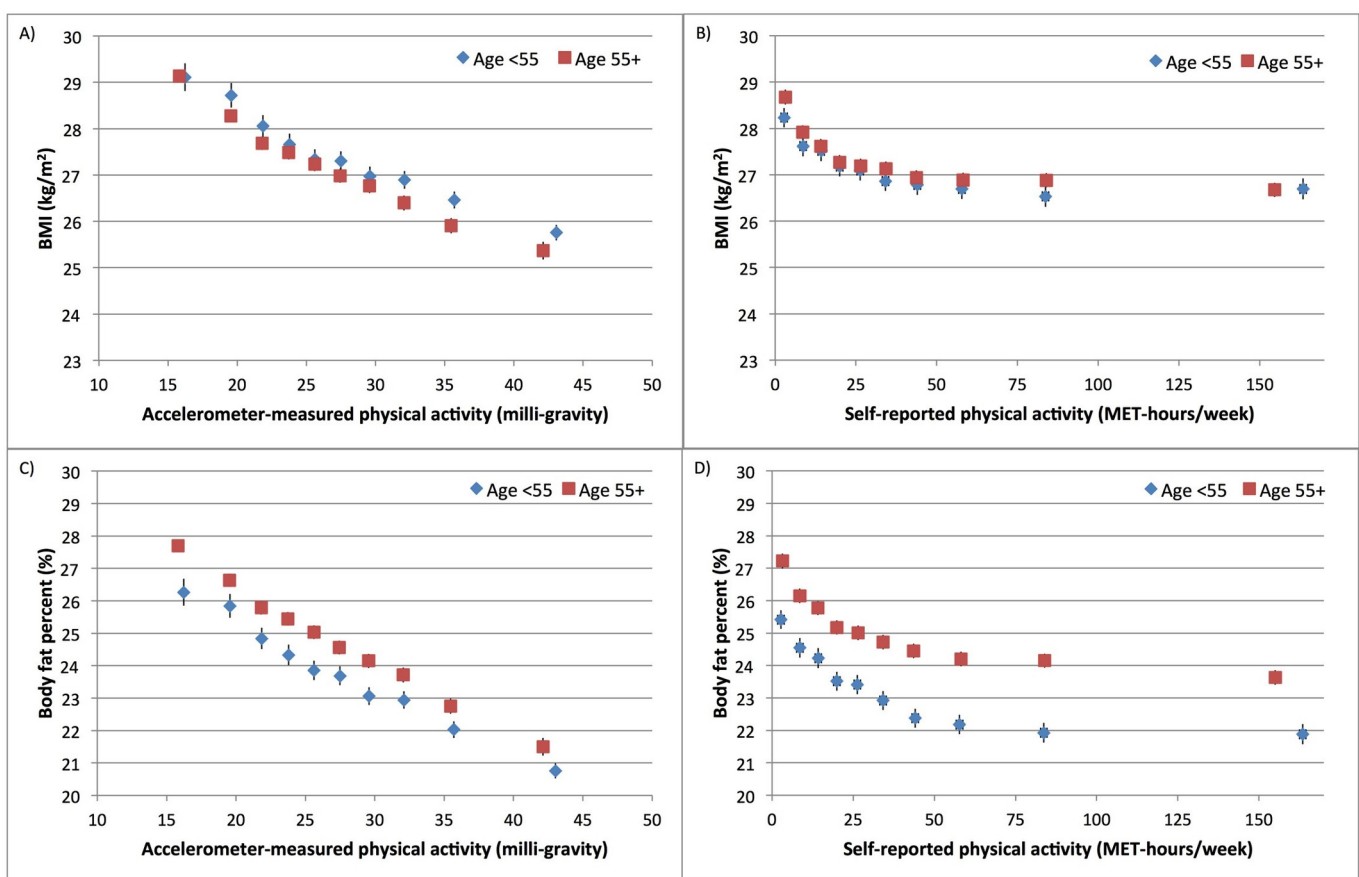

**Figure 3** Association of self-reported and accelerometer-measured physical activity with adiposity variables by age group in UK Biobank men. Association of physical activity measured by (A) accelerometer and (B) self-reported questionnaire with BMI. Association of physical activity measured by (C) accelerometer and (D) self-reported questionnaire with body fat per cent. Physical activity was grouped into 10ths. Adjusted geometric means (from linear regression models) for BMI and body fat per cent are plotted against the median value within each 10th of self-reported or accelerometer-measured physical activity. Adjusted geometric means are represented by diamonds for those under age 55 and squares for those ages 55 or older. These analyses are stratified by age at recruitment, region of recruitment and socioeconomic status, and are adjusted for educational qualifications, employment status, smoking status and alcohol intake frequency. The figure shows point estimates and 95% CIs. BMI, body mass index; MET, metabolic equivalent.

To our knowledge, the present study is the largest to date comparing accelerometer-measured and self-reported physical activity in relation to direct measures of body fat, although our results are consistent with prior, smaller studies that suggest a stronger association between adiposity and accelerometer-measured compared with self-reported physical activity.[18 20 23–26] This study was population based and recruited from 22 regions throughout the UK.[27] A major strength of this study is the availability of both accelerometer-measured physical activity and body fat by impedance in nearly 80 000 participants, together with data on body fat assessed by DXA in over 2400 participants. Additionally, the accelerometers used in this study were waterproof,[12] overcoming a limitation of prior studies where the devices had to be removed for water-based activities.[21]

While self-reported physical activity was available at baseline in these data, accelerometer-measured physical activity was assessed only 3–5 years after the end of recruitment, which raises the question of whether higher adiposity at baseline predicts lower physical activity levels[28] rather than physical activity determining adiposity. However, our analysis of accelerometer-measured physical activity in relation to DXA-measured body fat per cent, which was assessed within the same time frame as accelerometer-measured physical activity, showed similar results to the main analysis based on body fat per cent assessed by impedance at baseline. The accelerometer-measured physical activity variable available in UK Biobank at the time of these analyses cannot be directly compared with MET hours of self-reported physical activity. However, Willetts *et al* have recently developed physical activity phenotypes using a machine learning model with reference behaviours provided by data from a subset of participants who wore a camera along with the accelerometer.[29] Once these variables are made publicly available in UK Biobank, research using these metrics will facilitate the translation of study results into public health messages.

Other limitations include the lack of data on total energy intake for the whole cohort. Although accelerometer-determined physical activity is positively associated with per cent of lean muscle mass,[30] we did not consider

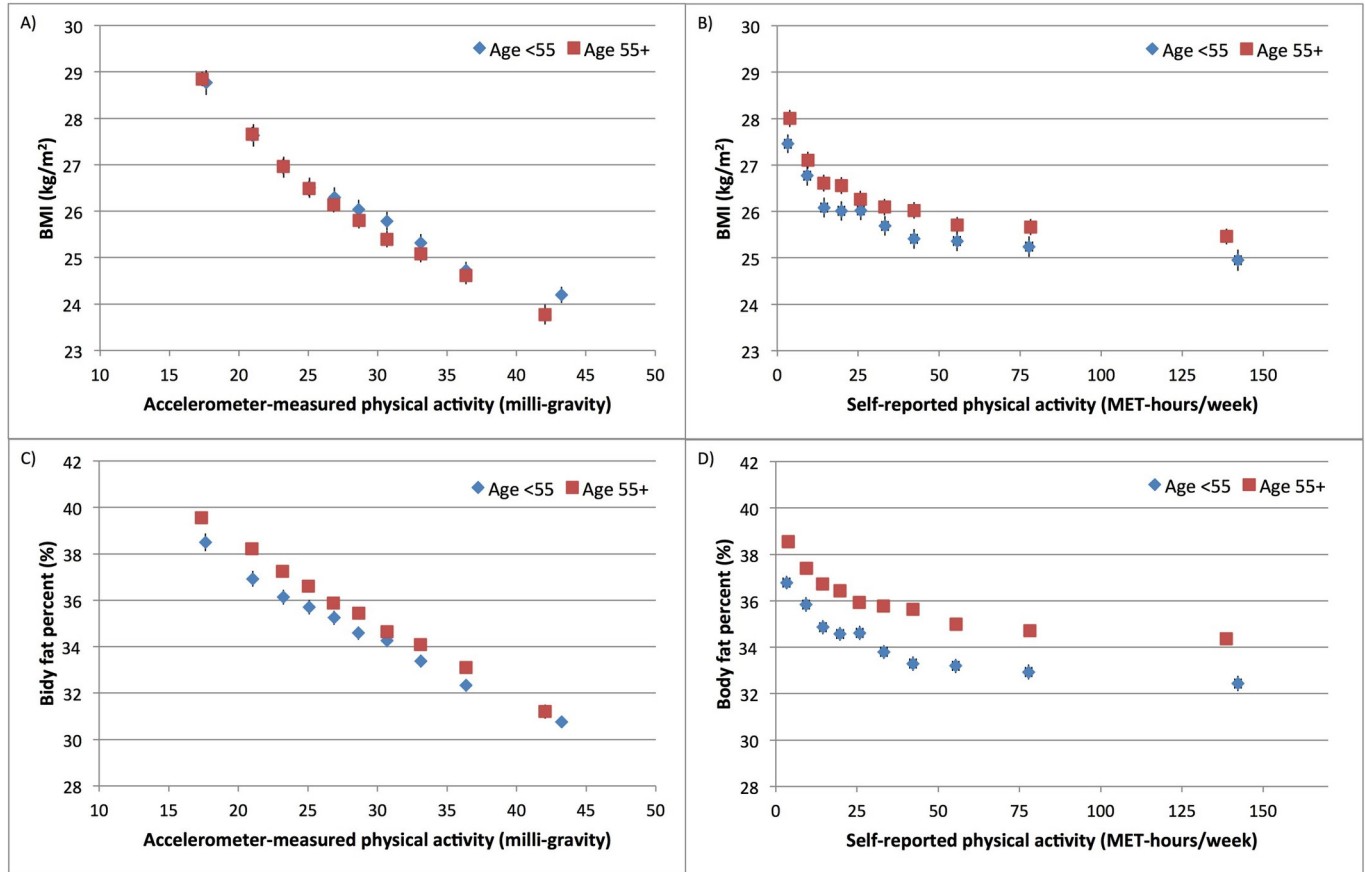

**Figure 4** Association of self-reported and accelerometer-measured physical activity with adiposity variables by age group in UK Biobank women. Association of physical activity measured by (A) accelerometer and (B) self-reported questionnaire with BMI. Association of physical activity measured by (C) accelerometer and (D) self-reported questionnaire with body fat per cent. Physical activity was grouped into 10ths. Adjusted geometric means (from linear regression models) for BMI and body fat per cent are plotted against the median value within each 10th of self-reported or accelerometer-measured physical activity. Adjusted geometric means are represented by diamonds for those under age 55 and squares for those ages 55 or older. These analyses are stratified by age at recruitment, region of recruitment and socioeconomic status, and are adjusted for educational qualifications, employment status, smoking status, alcohol intake frequency, parity, and hormone replacement therapy use. The figure shows point estimates and 95% CIs. BMI, body mass index; MET, metabolic equivalent.

this as a confounder in these analyses since we used data on direct measures of body fat per cent. Since accelerometer-measured time spent in sedentary activity was not available, we did not conduct analyses on sedentary activity. Due to the cross-sectional nature of this study, we cannot assess to what extent physical activity is causally related to adiposity. Highly active individuals may also be more likely to maintain appropriate target dietary energy intake, for example. Although the UK Biobank cohort is not representative of the general population in the UK, results of associations between exposures and health outcomes may be generalisable and would not necessarily require the study population to be representative if the biological basis of the exposure–disease relationship is shared.

In conclusion, our findings based on objective accelerometer data indicate a stronger relationship between physical activity and adiposity than previously thought. Comparisons of estimates of physical activity measured by questionnaire and by accelerometer suggest measurement error in self-reported physical activity, emphasising the need to incorporate objective measures of physical activity in future studies.

**Contributors** WG, TJK and GKR were responsible for study concept, design of the study, interpretation of the data and manuscript writing. WG had primary responsibility for statistical analysis and final content. All authors reviewed and approved the final manuscript.

**Funding** This work was supported by Cancer Research UK, grant number C570/A16491. WG is supported by the Clarendon Fund.

**Competing interests** None declared.

**Patient consent for publication** Not required.

**Ethics approval** UK Biobank has approval from the National Information Governance Board for Health & Social Care in England and Wales, the North West Multi-centre Research Ethics Committee and the Community Health Index Advisory Group in Scotland.

**Provenance and peer review** Not commissioned; externally peer reviewed.

**Data sharing statement** No additional data are available.

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
