## [Reviewer comments · BMJ Open]

ARTICLE DETAILS

TITLE (PROVISIONAL)	Accelerometer compared with questionnaire measures of physical activity in relation to body size and composition: a large cross-sectional analysis of UK Biobank
AUTHORS	Guo, Wenji; Key, Timothy; Reeves, Gillian

VERSION 1 – REVIEW

REVIEWER	Sophie Cassidy Newcastle University, UK
REVIEW RETURNED	21-Jun-2018

GENERAL COMMENTS	This large scale study investigates the correlation between self report and accelerometer derived physical activity and their associations with measures of adiposity. Strengths of the study are the large sample size, different measures of adiposity, and the results confirm previous observations of self report measurement error in those who are older and have a high BMI. I have some comments for the authors to address: 1-The major limitation with this study is that there was a time lag between self report and accelerometry. This has only been mentioned by the authors at the very end of the paper in the limitations section however it warrants mentioning earlier to make the reader aware that the associations with accelerometer and adiposity involve a time lag. Please comment on what the time lag actually was. 2-As the self report and accelerometer measures were not taken at the same time, I don't think it appropriate to correlate these measures. The paper should purely focus on the strength of associations with adiposity, not with each other. 3-From what I understand, DXA body fat results have only been released for around 5000 individuals. Please can this be made clearer and where this number is analysed then the 'n' is made clear-i.e. Figure 2. 4-As accelerometry is a main outcome, there should be more detail of how the authors analysed this data. They just reference the Doherty paper. Did the authors include all accelerometer files, or did they remove those with <3 valid days or those who did not have wear data in each one-hour period of the 24-hour cycle, as Doherty recommend? 5-Table 1: is the body fat % from BIA or DXA?
--

	6-There is little information about the accelerometer variable (mg) they included. Was this the average over the full-wear period? Is it just the day, or night too?
--	--

REVIEWER	Yi Chao Foong Barwon Health, Australia
REVIEW RETURNED	04-Jul-2018

GENERAL COMMENTS	General comments: Interesting, well-written paper examining the cross sectional relationship between PA (as measured by both objective and subjective means) and BMI/waist circumference/% body fat. Impressive numbers from the UK Biobank study. Major findings include: low correlation between subjective and objective measures of PA, PA was associated with lower adiposity. Findings are consistent throughout paper with dose-response relationship. Similar confirmatory findings to other studies done in the past in this area. Abstract and introduction: No suggestions for changes Methods:  -Participants: Why didn't roughly 80% of biobank participants not have accelerometer data? Was there any selection bias? -Self reported PA: Please find a reference for the use of 3.3/4/8 for MET cut offs for walking/moderate/vigorous PA -Body composition: Please explain how waist circumference was measured (which site was it measured from) -accelerometer: What do the measures actually translate to in real life i.e. how much PA is a milli-gravity equivalent to? -statistical analysis: What was the reasoning behind categorization instead of analyzing the data continuously? Results: Table 1:  -The most active men/women were also younger. Could age be a confounder? Was this adjusted for? -why have we analysed the data categorically instead of continuously? Figure 1/sup table 4:  -There is an impressively consistent dose response relationship and this should be commented on Discussion: Finding that objective assessment of PA did not correlate well with subjective measures is not new – please see Dyrstad et al Med Sci Sport Exerc 2014 Comparison of self-reported versus accelerometer measured PA Another strength not mentioned is that this was a community based study unlike some of the other studies which have relied on special population subsets Should also discuss the potential for lean muscle mass to be a confounding factor. See Foong et al JCSM 2016 Accelerometer-determined PA, muscle mass, and leg strength in community dwelling older adults Time spent in sedentary activity was also not considered/analysed, and should be acknowledged as a shortcoming
---

VERSION 1 – AUTHOR RESPONSE

Reviewer(s)' Comments to Author:

Reviewer: 1

Reviewer Name: Sophie Cassidy

Institution and Country: Newcastle University, UK

Please state any competing interests or state 'None declared': None declared

Please leave your comments for the authors below

This large scale study investigates the correlation between self report and accelerometer derived physical activity and their associations with measures of adiposity. Strengths of the study are the large sample size, different measures of adiposity, and the results confirm previous observations of self report measurement error in those who are older and have a high BMI.

I have some comments for the authors to address:

1-The major limitation with this study is that there was a time lag between self report and accelerometry. This has only been mentioned by the authors at the very end of the paper in the limitations section however it warrants mentioning earlier to make the reader aware that the associations with accelerometer and adiposity involve a time lag. Please comment on what the time lag actually was.

Response: We have now added that the accelerometer study occurred approximately 5.5 years after recruitment, when baseline physical activity was self-reported to the abstract, to the methods section under the subsection "Accelerometer-measured physical activity", and to the results section.

2-As the self report and accelerometer measures were not taken at the same time, I don't think it appropriate to correlate these measures. The paper should purely focus on the strength of associations with adiposity, not with each other.

Response: We understand the concern about the time lag and have now made the time lag between the self-reported and accelerometer measured physical activity clear throughout the entire manuscript. We believe it is useful to report these correlations given that the focus is more on the relative correlations across different characteristics (age, BMI, etc) rather than the overall correlation coefficient. We have also added the following to the discussion to make the time lag between the two physical activity measurements clear so that readers interpret the correlation coefficient with that information in mind: "the time lag between these two measurements of physical activity may have also contributed to a low overall correlation coefficient."

3-From what I understand, DXA body fat results have only been released for around 5000 individuals. Please can this be made clearer and where this number is analysed then the 'n' is made clear-i.e. Figure 2.

Response: There were 1,272 women included in the analysis and 1,185 men included in the analysis. We have now added this to the figure title. We have included the total number (n=2,457 participants) in the methods section under subsection "anthropometry and body composition."

4-As accelerometry is a main outcome, there should be more detail of how the authors analysed this data. They just reference the Doherty paper. Did the authors include all accelerometer files, or did they remove those with <3 valid days or those who did not have wear data in each one-hour period of the 24-hour cycle, as Doherty recommend?

Response: Yes, these details were provided in the methods section under subsection "study participants" in the original submission and shown in Supplementary Figure 1, which shows the exclusion criteria for the study. We excluded participants who did not have at least 72 hours (<3 days)

of accelerometer data and participants who did not have data in each one-hour period of the 24-hour cycle.

5-Table 1: is the body fat % from BIA or DXA?

Response: Body fat % is from BIA and we have added a footnote to clarify this.

6-There is little information about the accelerometer variable (mg) they included. Was this the average over the full-wear period? Is it just the day, or night too?

Response: Yes, under the subsection “accelerometer-measured physical activity” within the methods section, we have now specified that we used the “overall acceleration average” variable, which is data field 90012 in the UK Biobank dataset. We had previously stated that participants were instructed to wear the accelerometer “continuously for seven days.”

Reviewer: 2

Reviewer Name: Yi Chao Foong

Institution and Country: Barwon Health, Australia

Please state any competing interests or state ‘None declared’: None declared

Please leave your comments for the authors below

General comments:

Interesting, well-written paper examining the cross sectional relationship between PA (as measured by both objective and subjective means) and BMI/waist circumference/% body fat. Impressive numbers from the UK Biobank study. Major findings include: low correlation between subjective and objective measures of PA, PA was associated with lower adiposity. Findings are consistent throughout paper with dose-response relationship. Similar confirmatory findings to other studies done in the past in this area.

Abstract and introduction: No suggestions for changes

Methods:

-Participants: Why didn’t roughly 80% of biobank participants not have accelerometer data? Was there any selection bias?

Response: Out of the 236,519 UK Biobank participants who were approached to participate in the accelerometer study, 44.8% consented to join the study. Only participants who had a valid email address on file were sent an invitation to join the accelerometer study, and the invitations were sent via email. Aside from excluding participants without an email address, there was no selection bias in this invitation process, because the participant email addresses were chosen randomly (with the exception of the North West region which was avoided due to concerns of participant burden since this area had been used to trial other new projects).

UK Biobank overall is not representative of the sampling population in that there is evidence of a “healthy volunteer” selection bias (see Fry et al.) For the present analyses, we have restricted analyses to only participants who have both accelerometer and self-reported physical activity data.

Anna Fry, Thomas J Littlejohns, Cathie Sudlow, Nicola Doherty, Ligia Adamska, Tim Sprosen, Rory Collins, Naomi E Allen; Comparison of Sociodemographic and Health-Related Characteristics of UK Biobank Participants With Those of the General Population, *American Journal of Epidemiology*, Volume 186, Issue 9, 1 November 2017, Pages 1026–1034, <https://doi.org/10.1093/aje/kwx246>

-Self reported PA: Please find a reference for the use of 3.3/4/8 for MET cut offs for walking/moderate/vigorous PA

Response: We have added a reference for the IPAQ processing guidelines.

-Body composition: Please explain how waist circumference was measured (which site was it measured from)

Response: Waist circumference was measured at the level of the umbilicus. This has been added to the subsection “anthropometry and body composition” in the methods section.

-accelerometer: What do the measures actually translate to in real life i.e. how much PA is a milli-gravity equivalent to?

Response: The accelerometer-measured physical activity variable currently available in UK Biobank cannot be directly compared to MET hours of self-reported physical activity and is difficult to translate in practical “real life” terms. However, Willetts et al. have recently developed physical activity phenotypes using a machine learning model with reference behaviors provided by data from a subset of participants who wore a camera along with the accelerometer. Once these variables are made publicly available in UK Biobank, research using these metrics will facilitate the translation of study results into public health messages.

Matthew Willetts, Sven Hollowell, Louis Aslett, Chris Holmes, Aiden Doherty. Statistical machine learning of sleep and physical activity phenotypes from sensor data in 96,220 UK Biobank participants. *Sci Rep.* 2018 May 21;8(1):7961. doi: 10.1038/s41598-018-26174-1.

The above has now been added to the discussion on limitations of the study.

-statistical analysis: What was the reasoning behind categorization instead of analyzing the data continuously?

Response: We analyzed the data categorically instead of continuously because we did not want to make assumptions about linearity, especially since prior analyses demonstrated that the relationship between self-reported physical activity and adiposity is not linear.

Guo W, Bradbury KE, Reeves GK, et al. Physical activity in relation to body size and composition in women in UK Biobank. *Annals of Epidemiology* 2015;25(6):406-13 e6.

Additionally, we felt that categorization allowed us to communicate our findings more effectively by using figures that show the association between multiple categories of physical activity split by deciles and adiposity.

Results:

Table 1:

-The most active men/women were also younger. Could age be a confounder? Was this adjusted for?

Response: Yes, age was adjusted for as a confounder.

-why have we analysed the data categorically instead of continuously?

Response: Please see the detailed response above to the same question in the statistical analysis section.

Figure 1/sup table 4:

-There is an impressively consistent dose response relationship and this should be commented on

Response: We have added the following to the discussion: “There was a consistent dose-response relationship between physical activity and adiposity across the different measures of adiposity, which are highly correlated”, with a reference to a previous publication showing the correlation coefficients between the different measures of adiposity.

Discussion:

Finding that objective assessment of PA did not correlate well with subjective measures is not new – please see Dyrstad et al Med Sci Sport Exerc 2014 Comparison of self-reported versus accelerometer measured PA

Response: We have added this reference to the fourth paragraph of the discussion where we discuss the lower correlation between self-reported and accelerometer-measured physical activity in older participants.

Another strength not mentioned is that this was a community based study unlike some of the other studies which have relied on special population subsets

Response: We have added the following to the discussion: “This study was population-based and recruited from 22 regions throughout the UK.”

Should also discuss the potential for lean muscle mass to be a confounding factor. See Foong et al JCSM 2016 Accelerometer-determined PA, muscle mass, and leg strength in community dwelling older adults

Response: We have cited this paper and noted this point in the discussion.

Time spent in sedentary activity was also not considered/analysed, and should be acknowledged as a shortcoming

Response: This has been added to the paragraph on limitations in the discussion.

VERSION 2 – REVIEW

REVIEWER	Sophie Cassidy Newcastle University, UK
REVIEW RETURNED	25-Sep-2018

GENERAL COMMENTS	POINT 2: The authors justification for including a direct correlation between self-report and accelerometer measures is weak. There is a substantial time lag between the two measures and so they should not be compared in this way and should be removed from the paper, including table 2 + 3. There is merit in presenting the separate associations with body fat, and this should be the focus of the paper. CONCLUSIONS: ‘substantial measurement error’, this is very strong wording considering the time lag between both measures. The weaker associations of self-report and adiposity, indicate measurement error with self report PA, but this study cannot prove this. Pg 13, 3rd paragraph remove ‘lower adiposity to ‘adiposity’
--

REVIEWER	Yi Chao Foong Barwon Health, Australia
REVIEW RETURNED	24-Sep-2018

GENERAL COMMENTS	I am satisfied with the author’s replies and have no further concerns. Thank you for the opportunity to review the manuscript.
--

VERSION 2 – AUTHOR RESPONSE

Reviewer: 1

Reviewer Name: Sophie Cassidy

Institution and Country: Newcastle University, UK

Please state any competing interests or state 'None declared': None declared

POINT 2: The authors justification for including a direct correlation between self-report and accelerometer measures is weak. There is a substantial time lag between the two measures and so they should not be compared in this way and should be removed from the paper, including table 2 + 3. There is merit in presenting the separate associations with body fat, and this should be the focus of the paper.

Response: We agree that the main focus of the paper is on the separate associations of self-report and accelerometer physical activity with body fat and have, therefore, removed the correlations from the abstract. We would, however, like to include some mention of these correlations in the supplementary tables to help characterize the data for the reader. It is also important and relevant for all future studies in UK Biobank to have described the association between these two measures. To ensure that the reviewers concerns regarding the interpretation of such correlations are addressed, we have emphasized the time lag of approximately 5.5 years between the two measures in each section in which the correlation is mentioned, including the methods, results, and discussion sections. It is also clearly stated in the discussion section that "the time lag between these two measurements of physical activity may also have contributed to a low overall correlation coefficient."

CONCLUSIONS: 'substantial measurement error', this is very strong wording considering the time lag between both measures. The weaker associations of self-report and adiposity, indicate measurement error with self report PA, but this study cannot prove this.

Response: "demonstrate substantial measurement error" has now been removed

Pg 13, 3rd paragraph remove 'lower adiposity to 'adiposity'

Response: 'Lower adiposity' has been replaced with 'adiposity'

Reviewer: 2

Reviewer Name: Yi Chao Foong

Institution and Country: Barwon Health, Australia

Please state any competing interests or state 'None declared': none declared

I am satisfied with the author's replies and have no further concerns. Thank you for the opportunity to review the manuscript.

Response: Thank you.

VERSION 3 – REVIEW

REVIEWER	Sophie Cassidy Newcastle University
REVIEW RETURNED	26-Nov-2018
GENERAL COMMENTS	The authors have addressed all comments, and have dealt with my main concerns appropriately.